# Different Neural Activities for Actions and Language within the Shared Brain Regions: Evidence from Action and Verb Generation

**DOI:** 10.3390/bs12070243

**Published:** 2022-07-21

**Authors:** Zijian Wang, Qian Xi, Hong Zhang, Yalin Song, Shiqi Cao

**Affiliations:** 1School of Computer Science and Technology, Donghua University, Shanghai 200051, China; 2Department of Radiology, Shanghai East Hospital, Tongji University School of Medicine, Shanghai 200120, China; 96125007@sina.com; 3Department of Computer Science and Technology, Taiyuan Normal University, Taiyuan 030000, China; zhanghong8130@163.com; 4School of Software, Henan University, Kaifeng 475000, China; syl@vip.henu.edu.cn; 5Department of Orthopaedics, the Fourth Medical Center, Chinese PLA General Hospital, Beijing 100048, China; sq_cao@126.com; 6Department of Orthopaedics of TCM Clinical Unit, the Sixth Medical Center, Chinese PLA General Hospital, Beijing 100048, China

**Keywords:** fMRI, MVPA, action, language, supplementary motor area, inferior frontal gyrus

## Abstract

The Inferior Frontal Gyrus, Premotor Cortex and Inferior Parietal Lobe were suggested to be involved in action and language processing. However, the patterns of neural activities in the shared neural regions are still unclear. This study designed an fMRI experiment to analyze the neural activity associations between action and verb generation for object nouns. Using noun reading as a control task, we compared the differences and similarities of brain regions activated by action and verb generation. The results showed that the action generation task activated more in the dorsal Premotor Cortex (PMC), parts of the midline of PMC and the left Inferior Parietal Lobe (IPL) than the verb generation task. Subregions in the bilateral Supplementary Motor Area (SMA) and the left Inferior Frontal Gyrus (IFG) were found to be shared by action and verb generation. Then, mean activation level analysis and multi-voxel pattern analysis (MVPA) were performed in the overlapping activation regions of two generation tasks in the shared regions. The bilateral SMA and the left IFG were found to have overlapping activations with action and verb generation. All the shared regions were found to have different activation patterns, and the mean activation levels of the shared regions in the bilateral of SMA were significantly higher in the action generation. Based on the function of these brain regions, it can be inferred that the shared regions in the bilateral SMA and the left IFG process action and language generation in a task-specific and intention-specific manner, respectively.

## 1. Introduction

In recent years, the overlap and correlation of action and language cognitive processes has become one of the most critical research areas in cognitive neuroscience. The Action-Sentence Compatibility Effect (ACE) experiment [1] found that simultaneous language processing could modulate action planning and execution. Besides the findings from ACE, the interaction between language and motor systems has been found in neuroimage research. For example, functional magnetic resonance imaging (fMRI) studies have found that motion-related language could affect the neural activity in somatotopic regions of motor regions: the perception of verbs and sentences related to the hands, feet and mouth can trigger brain activation in corresponding regions of limbs [2,3,4]. Neuroimaging studies on the affordance of objects also found activation in the premotor cortex when participants read object nouns [5,6], suggesting that the motor codes related to object affordance are stored in the motor system, and the related motor codes are automatically activated while processing the object nouns. Neurophysiological studies also suggested that the action words caused neural activities close to the cortical representation of the body parts primarily used for carrying out the actions the verbs refer to [7,8].

All of the above findings suggest an interaction between movement and language processing in the cortical neural system. Some fMRI studies have also found more direct evidence that motor and language processing shared overlapping neural networks. For example, overlapping neural networks were found in the mental generation of actions and verbs [9,10]. Some other studies have found that the understanding of visually and verbally presented hand actions was processed in a polymodal manner in the shared neural substrates of the left Broca region [11]. Similar polymodal processing was found in understanding the action from the vocal-auditory or the gestural-visual stimuli [12].

However, the overlapping activation in fMRI does not mean that the same neurons are involved in different cognitive processes [13]. One plausible explanation is that the action and language tasks activate different patterns of neural activity in overlapping regions [14]. Previous studies, using univariate analysis, have found that the left inferior frontal lobe and premotor cortex are more engaged in language than action processing, while the left inferior parietal lobule is more activated for action than language tasks [10,14,15]. In addition to the univariate analysis, multi-voxel pattern analysis (MVPA) has been used more frequently in recent years to analyze the differences in neural activities between tasks in overlapping brain regions, which considers the activity of multiple voxels as a whole and can analyze the activity patterns of voxel activation after removing the average value [13,16,17,18,19]. Using MVPA on fMRI, Wurm et al. revealed that actions are encoded differently in the frontoparietal network for visually and written represented stimuli. However, the encoding patterns of different stimulus types are difficult to distinguish. In contrast, the motor representation of the left lateral posterior temporal cortex can be distinguished not only in motor categories, but also in stimulus types [20]. Zhang et al. found that the shared regions in the left frontal-parietal cortex activated similarly for action and language phonological processing. However, the shared regions in the frontal-parietal network adjust to phonological or action modal processing based on the requirements of tasks by using MVPA [14]. MVPA also has been employed to reveal the separation between the language and action inner simulations involving the same action semantics in the primary motor cortex and the premotor cortex [21].

Although previous studies have reported that the action and action-related language generation shared the frontal-parietal network [10], there is still a lack of research to reveal differences in activation patterns between actual action and language generation tasks in shared brain regions. The findings on this issue might help us better understand the differences in neural processing associated with action and language generation. Understanding the neural processing differences between action and language generation might facilitate the understanding of action impairment disorders such as Parkinson’s disease [22,23] and apraxia [24,25], in which patients are found to have difficulty in actual action and action verb generation [26,27,28]. In particular, when generated actions and language contain the same intention, the different patterns of neural activities in the shared brain region might suggest that this region processes the same semantic information and is involved in task-specific processing. Considering that action-related language involves many aspects, including phonology, semantics and grammar, we hope to provide critical conclusions by analyzing the common brain activation regions and activity patterns during action and verb generation in this work.

Previous studies have found that the inferior frontal gyrus (IFG) and dorsal premotor cortex (dPMC) are involved in action and language processing. IFG is involved in semantic understanding and semantic integration in language processing [29,30] and motor imitation and observation processing [31,32]. The dPMC was associated with action execution, action simulation and verbalization [30]. According to the above findings, action and verb generation tasks may have common activation regions in the IFG and PMCd. The neural activities in the two tasks are analyzed after localizing the shared regions.

In this study, we conducted an fMRI experiment of action and verb generation for object noun stimuli. We used conjunction analysis to locate the shared activation regions of the two tasks. We hypothesized that action and verb generation might have different patterns of neural activity in the shared region. To analyze whether their patterns were dissociated, we compared the mean activation levels of the two tasks in the shared area and analyzed whether their activity patterns were separable using MVPA. In the mean activation level comparison, we compared whether the estimated weight parameters of the two tasks in the shared region were similar in the general linear model (GLM). The dissociation means that the intensity of neural activities of action and verb generation is significantly different. In the MVPA analysis, we used the machine learning method to train and classify the normalized neural activities of the two tasks in the shared region. The high classification accuracy indicated that the action and verb generation cognitive processing activities were significantly different after removing the mean activity levels. We predicted that action and verb generation tasks’ activation levels and activity patterns would significantly differ in the shared regions.

## 2. Materials and Methods

### 2.1. Participants

A total of 21 healthy participants participated in this experiment, including 13 males and 8 females, with an average age of 22.7 years old and an age standard deviation of 1.23 years old. All the participants were right-handed, had no history of mental illness or impairment, and were native Chinese speakers. They are all undergraduate or graduate students, whose years of education range from 14 to 18 years. Each of them signed an informed consent form and were paid the same amount after the trial. The experiment was conducted in the Jiading Campus of Tongji University and was approved by the Ethics Committee of Tongji University.

### 2.2. Visual Stimuli

74 object nouns were selected from the Bank of Standardized Behaviorists (BOSS) and the internet. All written words of the nouns with nouns were scaled to 256 × 256 pixels with a light grey background. All the stimuli were rated on tests of action consistency and familiarity before the fMRI experiment. Eleven participants took part in the rating study, where they were asked to perform two tasks for each object noun: first, to select the most appropriate action to manipulate the object; second, to score the degree of familiarity with an object (0–7, ranging from unfamiliar to very familiar). These participants were all undergraduate or graduate students, between the ages of 20 and 24, and had no history of mental illness or impairment, similar to the participants in the fMRI experiment.

The objects with high action consistency and familiarity were used in the fMRI experiment. The action consistency was calculated according to Equation (1):(1)$$action\;consistency_{i}=\frac{N_{i,1}}{N_{i,2}}$$

$N_{1}$ is the number of the most common related action for object noun *i*. $N_{2}$ is the number of the second most common related action for object noun *i*. The action consistency for noun *i* is the ratio of $N_{1}$ to $N_{2}$. The familiarity score is the average score of the participant’s familiarity with the noun words. From the results, we selected 24 nouns of objects as stimuli. The average action consistency was 9.57 (SD = 0.24, more than 87% of answers were identical), and the average familiarity was 5.44 (SD = 0.17). Every 8 of the 24 objects were associated with three different hand movements: holding, pinching and pressing. Selected stimuli are shown in Figure 1.

### 2.3. fMRI Acquisition and Design

In this study, a 3T GE MRI scanner (GE Healthcare, Waukehsa, WI, USA) was used for data acquisition. The gradient Echo-planar imaging (EPI) sequence was used to scan fMRI images, and the scanning parameters were set as follows: repetition time (TR) = 2000 ms, echo time (TE) = 30 ms, scanning field of view (FOV) = 220 × 220 mm^2^, 37 slices, slice thickness = 3.4375 mm, voxel size is 3 × 3 × 3.4375 mm^3^. A total of 324 functional images were collected in two scanning sequences for each participant. In the experiment, we also collected the T1-structural image of each participant by a 3D FSPGR sequence [TR = 8.5 ms, TE = 3.2 ms, FOV = 256 × 256 mm, slice thickness = 1 mm], and the voxel size was 1 × 1 × 1 mm^3^.

When the participants were lying in the MRI scanner, two sponges on the two sides were used to fix the participants’ heads to reduce head movement. Participants were reminded before the experiment to keep their heads still during the whole scan. We conducted a dummy scan for 10 s before each scan sequence to make the MRI scanner reach a stable state, and a scan for about 5 min without participants before each experiment to test the state of the MRI scanner. A block design was used in the experiment. There are three tasks in the experiment. The first task is the action generation task for an object noun (NA). Participants were asked to really generate an appropriate action according to the object nouns. The second task was the action verb generation for an object noun (NV). Participants were asked to generate an appropriate action verb silently according to the object nouns. The third task was the noun reading task (NR), which was the control task of NA and NV. Participants were asked to name the object picture silently. Rest blocks were interspersed between two consecutive task blocks, in which participants were presented with a blank screen with a “+” symbol in the center. In this experiment, it was difficult to record the behavioral data in real time in the MRI. Therefore, immediately after the scan, participants were presented with the noun words and were asked to recall the actions and verbs that were generated in the experiment for each noun. Each noun word was presented only once, in the order in which they were first presented under the MRI. All nouns were scaled to the same size as those seen in the MRI scanner. The experimental program was written and presented by Presentation Software v0.71 (NeuroBehavioral Systems, CA, USA).

The orders of tasks and blocks in each run were counter-balanced. There were two runs in the experimental session. One run included twelve task blocks and twelve rest blocks. Action generation, verb generation and naming tasks each had four blocks. A visual prompt for the coming task was presented for 2 s before each task block, for example, “Please execute an action related to the presented object noun” (Chinese, “请执行物体名称相关动作”) for the NA task. Each task block comprised six trials. The stimulus presented for 1500 ms after a fixation cross for 1000 ms in a trail. One rest block lasted for 10 s. The experiment design is shown in Figure 2.

### 2.4. Behavioral Analysis

In the action and verb generation tasks, responses in the recall task were recorded immediately after scanning. We calculate the accuracy of the subjects’ responses to each stimulus and determine whether the subjects’ responses are the same as expected. Then, the paired *t*-test was used to check whether participants responded differently in action and verb generation for the same noun words.

### 2.5. fMRI Data Analysis

The standard preprocessing was performed on all functional images using SPM12 (Statistical Parametric Mapping, Wellcome Department of Cognitive Neurology, London, UK). BOLD images in the same run were realigned to the first slice. The head motion parameters were estimated in the realignment. The structural images were co-registered with the mean functional images. The structural image was segmented into different tissue types: grey matter, white matter and cerebrospinal fluid. The functional images were transformed to the Montreal Neurological Institute (MNI) space using the estimated transformation parameters in segmentation. Finally, the functional images were smoothed using an 8 mm FWHM kernel.

The general linear model (GLM) was used for individual statistical analysis. We modeled the tasks of NA, NV, NR and resting as independent regression variables. Head motion estimation was incorporated into GLM as an additional regression variable. The slow signal drift is eliminated by using a 1/128 Hz cutoff high-pass filter. The sequence correlations between the functional images were modeled using the AR(1) model. Four contrasts were performed in first-level analysis: NA > NR, NV > NR, NA > NV and NV > NA. The contrast images of each participant were used for group-level analysis, using a one-sample *t*-test. The threshold values of the activation maps were *p* < 0.001 at the voxel level and *p* < 0.05 at the cluster level with family-wise error (FWE) correction.

We located brain regions that were commonly activated in NA and NV tasks by conjunction analysis. In conjunction analysis, we extracted the voxels that passed the statistical threshold of NA and NV tasks and found that these voxels were activated under both task conditions. These voxels were overlapping regions shared by NA and NV. Then, these regions were separated by the anatomical brain regions they were located in. For example, shared regions in the left Inferior Frontal Gyrus (IFG) were labeled “shared-LIFG”. We employed paired sample *t*-test to compare the mean activation levels of NA and NV tasks in the common activated areas.

### 2.6. Multi-Voxel Pattern Analysis

MVPA using a linear support vector machine (SVM) was conducted in the labeled sub-regions to test whether action and verb generation for the same stimuli had similar activation patterns. The voxel signal series in each ROI were normalized along the temporal and spatial dimension before classification, by Equation (2):(2)$$b_{j}^{i}=\frac{\frac{a_{j}^{i}-\mu_{j}}{\sigma_{j}}-\mu^{i}}{\sigma^{i}}$$

Here, the $a_{j}^{i}$ is the raw voxel value for the jth voxel at the ith timestep, and the $\mu_{j}$ and $\sigma_{j}$ are the average and standard deviation values of the jth voxel timeseries, which means that the normalization along the temporal dimension is prior to that along the spatial dimension. The $\mu^{i}\;$ and $\sigma^{i}$ are the average and standard deviation values of normalized voxel activities in the ROI at the ith timepoint. After two-step normalization, the mean activities of voxels in each ROI were normalized to zero. The MVPA results based on the normalized activity show whether each ROI has a similar pattern after stripping out the average activity in different tasks.

We put each TR into the MVPA as a data sample, and for each of the two compared tasks, there were 128 data samples per participant. We used 8-fold cross-validation to calculate the average classification accuracy, in which 7/8 data were used for model training and the other 1/8 data were used for testing in each of the 8 cross-validations. Finally, the average classification accuracy of all the subjects in each ROI was taken as the classification accuracy of the two tasks. A one-sample *t*-test was performed for comparing the classification accuracy to 50% (chance probability of two classes). If it was significantly greater than 50%, the two activity patterns were considered different; otherwise, they were similar. The linear SVM was implemented using sklearn 0.23.0 (https://scikit-learn.org, accessed on 1 December 2020) with default hyperparameters preset by sklearn.

## 3. Results

### 3.1. Behavioral Results

The mean accuracy of the recall responses of NA and NV compared with the predefined responses was 95.31% (SD = 0.034) and 94.27% (SD = 0.050), respectively. We performed paired samples *t*-tests on the two tasks and found no significant difference between these two tasks (*t*(20) = 0.939, *p* = 0.359). These results showed that the responses were consistent with the predefined actions and not affected by tasks or stimuli.

### 3.2. Brain Activation

Activation for NA > NR was found in the bilateral Supplementary Motor Area (SMA), the left Superior Frontal Gyrus (SFG) and the Inferior Frontal Gyrus (IFG) (Figure 3a; Table 1). Activation for NV > NR was found in the bilateral Supplementary Motor Area and the left IFG (Figure 3b; Table 2). Activation for NA > NV was found in the left Precentral Gyrus, the bilateral SMA and the left Supramarginal Gyrus (Figure 3c; Table 3). The activation in the SMA was mostly located in the pre-SMA. However, no activation was found for NV > NA. According to these results, it can be preliminarily found that the brain activation is mainly biased to the left hemisphere in both action and verb generation tasks. The SMA and IFG were activated by both action and verb generation tasks. However, the activation of action generation in the Premotor Gyrus and Supramarginal Gyrus was significantly stronger than that of verb generation.

Although the comparison of NA > NV at the voxel level found the Premotor Gyrus, SMA and Supramarginal Gyrus with stronger activation for action generation than verb generation, in order to further distinguish the difference in activation between the two tasks in the integral activity pattern, we need to locate the overlapping activation regions and compare the voxel activity patterns in these regions by conjunction analysis. Conjunction analysis was performed between the NA > NR and NV > NR. Overlapped activation of NA > NR and NV > NR was found in the bilateral SMA and the left IFG, shown in Figure 4 and Table 4. Activation of the SMA was mostly found in the pre-SMA.

### 3.3. Analysis of Voxel Activity Pattern

#### 3.3.1. Regions of Interest

We segmented shared regions according to the cortex regions of the overlapping activation of action and verb generation and selected them as regions of interest. The shared regions were found in the left Supplementary Motor Area (SMA), the right SMA and the left IFG, each named shared-LSMA (78 voxels), shared-RSMA (38 voxels) and shared-LIFG (29 voxels), respectively. These shared regions are shown in Figure 5. In each ROI, we extracted the voxel time series under the action and verb generation tasks and compared the differences between the two tasks in the common activation regions by means activation level analysis and multi-voxel pattern analysis, respectively.

#### 3.3.2. Mean Activation Level Analysis

Mean activation levels were compared between NA > NR and NV > NR tasks in all the shared areas (Figure 6). Mean activation levels for NA > NR were significantly higher than NV > NR in shared-LSMA (*t*(20) = 2.97, *p* = 0.004, Bonferroni corrected) and shared-RSMA (*t*(20) = 3.65, *p* = 0.001, Bonferroni corrected). However, these two contrasts did not significantly differ in shared-LIFG (*t*(20) = 1.54, *p* = 0.070). It can be found that although the *p* value in shared-LIFG is also close to 0.05, the difference in the mean activation level between action and verb generation is smaller than that in the other two ROIs. Since the generated actions were consistent with the meaning of the verb, this might indicate that the overlapping brain regions of the left IFG process the action intention information more than those of the SMA.

#### 3.3.3. MVPA Results

We examined whether voxel-wise activity patterns could differentiate the NA and NV tasks in each shared ROI in MVPA. Classification accuracy is shown in Figure 7 and Table 5. The classification accuracy in all ROIs was significantly greater than the chance-level accuracy of 50%, revealed by paired *t*-tests (*p* < 0.05, Bonferroni corrected). These results indicated that the MVPA results of shared-LSMA and shared-RSMA were consistent with the mean activation level analysis results, showing significant differences between action and verb generation. As for shared-LIFG, its classification accuracy was still lower than the two ROIs of SMA, but its activity pattern was significantly distinguishable, which was not consistent with the mean activation level analysis.

*t* and *p* values were obtained from the paired-sample *t*-tests between the average accuracy and the chance-level accuracy of 50%. All of the accuracies survived Bonferroni correction.

## 4. Discussions

Using well-designed fMRI experiments and conjunction analysis, we found that the activation intensity of the left Precentral Gyrus, the left SMA and the left Supramarginal Gyrus in the action generation task was significantly higher than that of verb generation. We also found the common neural substrates of action and verb generation for an object noun in the bilateral SMA and the left IFG. Furthermore, using the mean activation level and the MVPA analysis, we provided the first fMRI evidence that the voxel activity patterns of action and action verb generation with the same meaning for object nouns within the shared regions in the bilateral SMA were different. However, the activation patterns of the two tasks in the shared-LIFG were slightly different.

### 4.1. Shared and Specific Neural Substrates for Action and Verb Generation

The left SupraMarginal, located in the left Inferior Parietal Lobe (IPL), was only involved in the action generation task resulting from NA > NV. The IPL was suggested to be related to encoding the abstract information of object-directed actions [33]. It also plays a vital role in action organization and understanding the intention of others’ actions [34]. IPL was also proposed to be involved in verb generation versus visually presented objects or faces [6,30,35]. It was found that the action verb and the corresponding action video observation can lead to the activation of the same neuron in the Posterior Parietal Cortex (PPC), which is closer to the IPL [36]. According to the above research on action and verb processing functions in the IPL, it can be inferred that the IPL is closely related to action execution, understanding and intention coding, and participates in verb retrieval and observation. It seems to contradict the results of this study that the verb generation task is not activated in the IPL. However, most of the other experiments that have found the activation of verb generation in the IPL have used object or face observation as control experiments, while this study used object noun reading as a control experiment. The IPL was proposed to decode different categories of noun information more accurately than verbs [37], which shows that the IPL is more involved in retrieving nouns than verbs. Therefore, the activation of verb generation in the IPL was probably weakened by noun reading, and no activation was found in this study. However, action generation was found significantly activated in the left IPL, perhaps because the action generation required more neural activity than verb generation. This result suggests that the action generation task may need more significant activity in the left IPL than the verb generation and noun reading tasks, consistent with the finding that the left parietal lobe is more involved in action processing than language processing [14].

The bilateral Supplementary Motor Area, especially the pre-SMA, was found in all contrasts: NA > NR, NV > NR and NA > NV, as well as in the conjunction analysis. The SMA is located in the dPMC and the midline surface of the PMC. It was found to be involved in the planning, initiation and control of actions [38,39,40] and in the inner speech and context integration of language functions [41]. In this work, two types of sub-regions were located in the bilateral SMA according to the results of brain activation and conjunction analysis: the sub-region specified by action generation and the region shared by action and verb generation. The sub-region specified by action generation located in both dPMC and the midline surface of the PMC might be only involved in the control of actions. However, the shared region is located only in the pre-SMA, which might be involved in both action control and action language processing. Previous works have found that pre-SMA is involved in language processing and production [41,42]. Based on the above analysis, we can obtain a more accurate inference: the pre-SMA involves action control and language processing.

The conjunction analysis of NA > NR and NV > NR also showed that the left Inferior Frontal Gyrus was involved in the action and verb generation for object nouns. The IFG is suggested to be involved in the execution, observation and imitation of actions [30,31,43], as well as the semantic processing, semantic integration and generation of words [29,44,45]. However, no significant difference was found in the left IFG in the NA > NV, which showed that the left IFG was also shared by action and action-related verb generation for object nouns. This result is consistent with previous findings [30]. However, demonstrating that one brain region is involved in two tasks at once is not clear enough to infer what is going on in this brain region. Therefore, we will analyze the patterns of neural activity and discuss how these shared areas are involved in action and verb generation.

The results of brain activation were also consistent with the psycholinguistic model. Many researchers have claimed that the neural basis of language processing is widely distributed [46,47]. This research suggested that action naming and noun naming were coprocessed by the dorsal and ventral stream tracts. Our results support the distributed psycholinguistic model and further suggest that the distributed brain regions for verb processing were overlapping with the regions for action processing. This might suggest that the underlying meanings and intentions of the generated actions and verbs were processed by the similar neural substrates.

### 4.2. Different Neural Activity Patterns within the Shared Neural Areas

We analyzed the mean activation levels and multi-voxel patterns of the action and verb generation tasks in three shared ROIs. The results of the two analyses in the shared-LSMA and shared-RSMA were consistent, which showed different neural activity patterns in the two regions and a significantly higher activation level for action generation. In this study, the behavioral results of the participants’ actions and verb generation tasks were highly consistent with the predefined responses, indicating that the intention of generated actions and verbs was the same in both tasks. Thus, the difference in neural activity between action and verb generation with the same intention should be due to how the task is performed: actual execution and inner speech. Actual execution requires action control, and inner speech requires more lexical-semantic processing [48]. The different neural activity in the shared-LSMA and shared-RSMA suggest that these shared regions may be involved in action control and lexical-semantic processing in action and verb generation, respectively. This conclusion supports previous findings that the bilateral SMA is involved in motor control and language processing [38,40]. Since the shared regions in the bilateral pre-SMA were activated in both action and verb generation for tool nouns, it may be involved in encoding action intention. Moreover, in the comparison of mean activation levels, activation levels of these regions were significantly higher in action generation than in verb generation. A possible explanation for this might be that these regions are more inclined to action control than lexical-semantic processing.

Surprisingly, the shared-LIFG was found to have a separable pattern of activity in MVPA, but the mean activation levels for action generation tasks were not significantly higher than for verb generation. The left IFG was suggested to play an essential role in semantic retrieval, language generation and action intention encoding [49,50,51]. The insignificant difference is likely to be influenced by the similar intention of the generated actions and verbs. However, this area should not only have the function of intention encoding in action and verb generation. Otherwise, their activity patterns in MVPA would not be different. One possible explanation is that this region, like the shared region of SMA, may be involved in action control in action generation and lexical-semantic processing in verb generation. However, if action control and lexical-semantic processing dominated the shared-LIFG, the results of mean activation level analysis and MVPA should be similar to those of the shared-LSMA and shared-RSMA. Therefore, it could be inferred that the involvement of these two functions in this region may not be more necessary than that of intention encoding. That should be why only MVPA found the different activation patterns in this region, which is more sensitive than mean activation level analysis [16].

Combining the above results, we can deduce that the shared regions in the bilateral SMA and the left IFG are involved in intention encoding, action control and lexical-semantic processing. These regions encoded the action intentions together in action and verb generation, leading the two tasks to activate these regions commonly. The shared regions in the bilateral SMA were mainly modulated by the task context, in which the action control and lexical-semantic processing dominated these regions. However, the shared region in the left IFG was modulated by intention context and influenced by the task context to a lesser degree, which could be only detected by MVPA. The severity of Parkinson’s disease was found to be related to brain activation of the left SMA and the left IFG, and the neural activity of the left SMA is also related to anticipatory postural adjustments of Parkinson’s patients [22,23]. Apraxia of limbs is suggested to be related to SMA lesions [24]. IFG lesions were suggested to result in the combined defects of apraxia and aphasia, especially the impairment of language and motor intention processing [25]. These finding could be matched by studies of patients with Parkinson’s disease and apraxia. The results of this work could also help research on the early diagnosis and retreatment of Parkinson’s disease and apraxia, which might provide valuable results on brain activity for designing biomarkers, although previous studies have found overlapping neural networks for action and language processing. Our works further reveal that the same neural regions in the bilateral SMA and the left IFG process action and language generation in a task-specific and intention-specific manner, respectively.

### 4.3. Limitations

There were some limitations in this study that may affect the results of the experiment. The first limitation was the lack of behavioral data collection under fMRI. Due to the lack of effective behavioral response recording methods, the experiment could not record participants’ actions and silently generated verbs in real time. Therefore, we asked participants to recall the generated actions and verbs immediately after the MRI scanning. However, it is difficult to exclude the trail records with inconsistent behaviors due to the lack of specific responses recorded for each trail. The second limitation is that the *p* value was not corrected in GLM analysis. This is because only 21 subjects participated in this experiment, and the activation intensity of the verb generation experiment was weak, so the corrected activation region was too small. Therefore, we only used the cluster-level FWE correction in GLM analysis. However, since we still limited the voxel level to *p* < 0.001, this could cause a minor impact on the results of mean activation level and activation pattern analysis in overlapping regions. Finally, since the pre-experiment was conducted for free, we recruited only 11 participants to score the familiarity and action consistency of object nouns. Therefore, these limitations might cause a small impact on the experimental results. We will solve these limitations through better experimental design and participant recruitment in future experiments.

The parameters of image acquisition and processing also could affect the results in this work. The readout bandwidth (BW) and echo spacing (ES) values could significantly influence the temporal stability and image quality of EPI-fMRI time series [52,53]. The low BW could improve both the signal-to-noise ratio of EPI-fMRI time series and the temporal stability of functional acquisitions. The minimum ES values could not be optimal when the MR scanner system is characterized by gradients with low performances and suboptimal EPI sequence calibration. However, the BW and ES values in the GE MRI scanner were nonadjustable, which could affect the quality of EPI-fMRI time series. In the fMRI acquisition, we used a suboptimal anisotropic voxel. The anisotropic voxel was found to generate more volume deviation than the isotropic voxel. In fMRI data preprocessing, we used a kernel of 8 mm FWHM in smoothing and eliminated the slow signal drift using a 1/128 Hz cutoff high-pass filter. We used the 8 mm FWHM for combining more activated voxels into activated clusters. However, the 8 mm FWHM could lead to some false activation regions [54], and the high-pass filter was used to correct the drifts caused by scanner instabilities and other sources. The 1/128 Hz cutoff high-pass filter was selected by referring to previous related studies. However, we did not compare the performances of different filters. The selection of the filter could influence the removal of valuable signals or noise. Therefore, these limitations on data acquisition and preprocessing methods may also have an impact on the results.

## 5. Conclusions

In this work, we conducted an fMRI experiment comparing the difference in brain activities for action and verb generation to that of the same object nouns. The results of activation level analysis suggested that the action generation task might need more significant activity in the left IPL than the verb generation and noun reading tasks. The left IPL could participate in the action’s execution, understanding and intention coding, and verb retrieval and observation. The bilateral SMA and the left IFG were found to involved in action controls and language processing. The results in activation pattern analysis gave more conclusions. These regions encoded the action intentions together in action and verb generation. The shared regions of bilateral SMA are dominated by task context, including action control or lexical semantic processing. However, the shared region of the left IFG region is regulated by the intention context and is less affected by the task context, which can only be detected by MVPA. The brain activity in SMA and the IFG was found to be related to Parkinson’s disease and apraxia. In addition to supporting the neural mechanism of Parkinson’s disease and apraxia, our findings also provided evidence on neurosciences for the early diagnosis, prognosis evaluation and rehabilitation program design of these diseases. For example, for patients with Parkinson’s disease or apraxia, the neural activity of SMA and IFG could be abnormal in the early stages. In the future, the results of this work could facilitate research on biomarkers of these diseases and promote the early diagnosis and treatment of motor function impairment.

## Figures and Tables

**Figure 1 behavsci-12-00243-f001:**
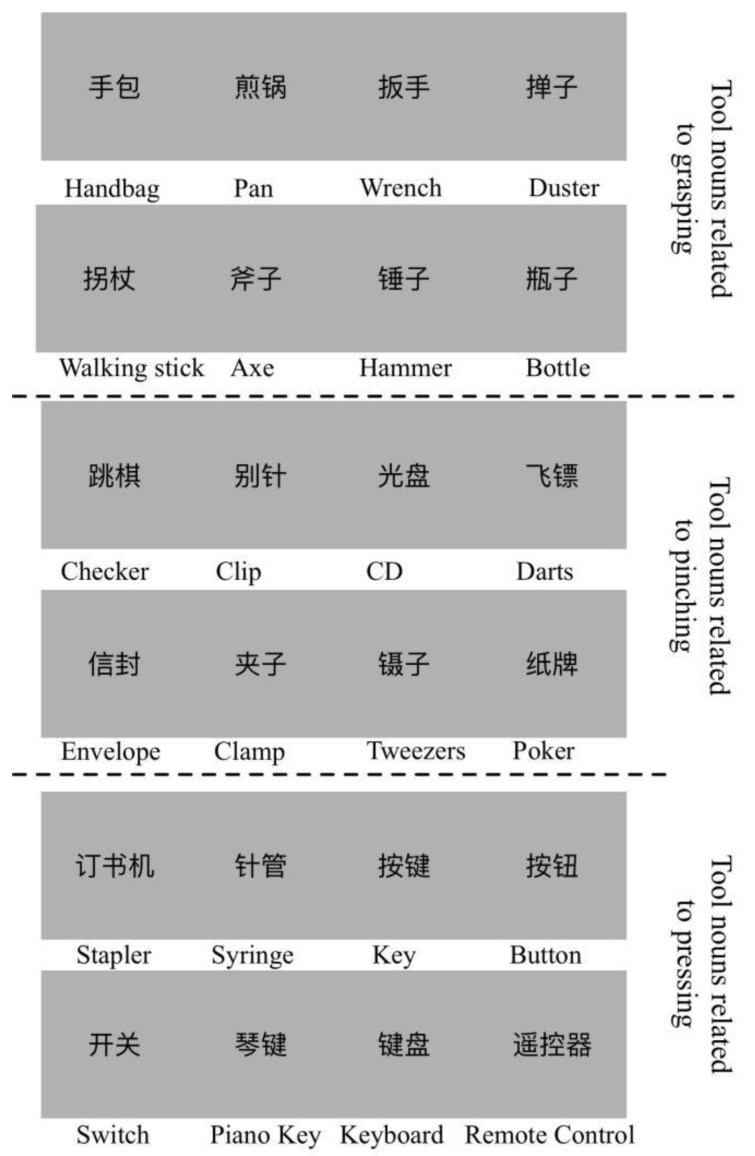
The selected stimuli used in the fMRI experiment with high action consistency and familiarity. Twenty-four object nouns were selected, and the written noun words were scaled to 256 × 256 pixels with a light grey background. The corresponding English words are below each stimulus. For example, the “手包” in Chinese is the “handbag”. Each of the eight object nouns were highly correlated with grasping, pinching or pressing.

**Figure 2 behavsci-12-00243-f002:**
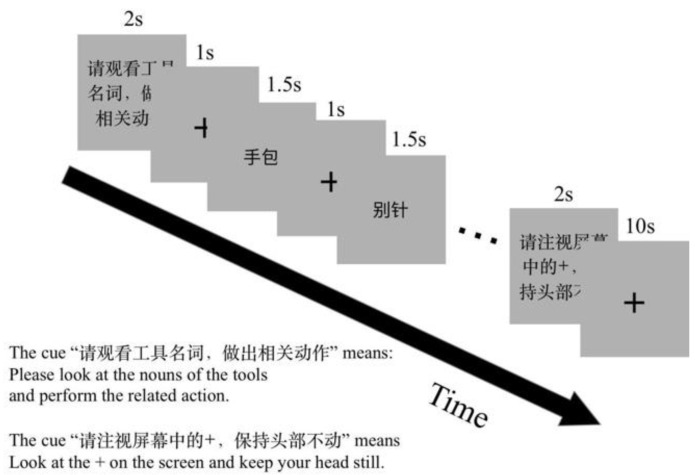
The fMRI experiment paradigm design. There were two scan runs in this experiment. One run comprises three tasks: NR, to silently read an object noun; NA, to perform a hand gesture to an object noun; NV, to silently generate a verb to an object noun. Before the task block, a visual cue was presented to prompt the task.

**Figure 3 behavsci-12-00243-f003:**
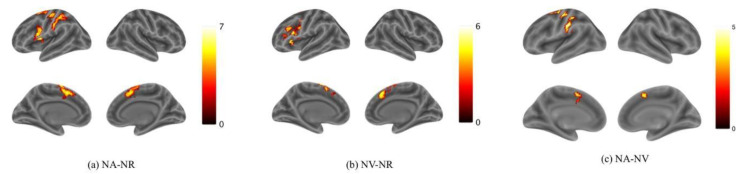
Activation maps for (**a**) NA > NR, (**b**) NV > NR and (**c**) NA > NV. Color bars represent *t* values. The threshold values of the activation maps were *p* < 0.001 at the voxel level and *p* < 0.05 at the cluster level with family-wise error (FWE) correction.

**Figure 4 behavsci-12-00243-f004:**
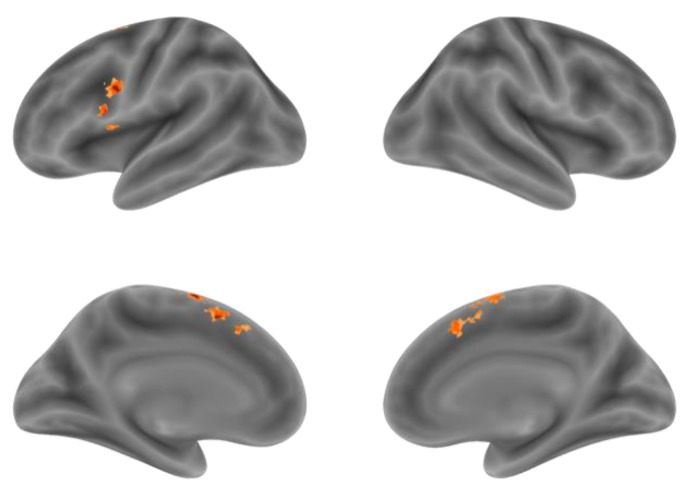
The overlapping activation regions of NA > NR and NV > NR in the bilateral SMA and the left IFG.

**Figure 5 behavsci-12-00243-f005:**
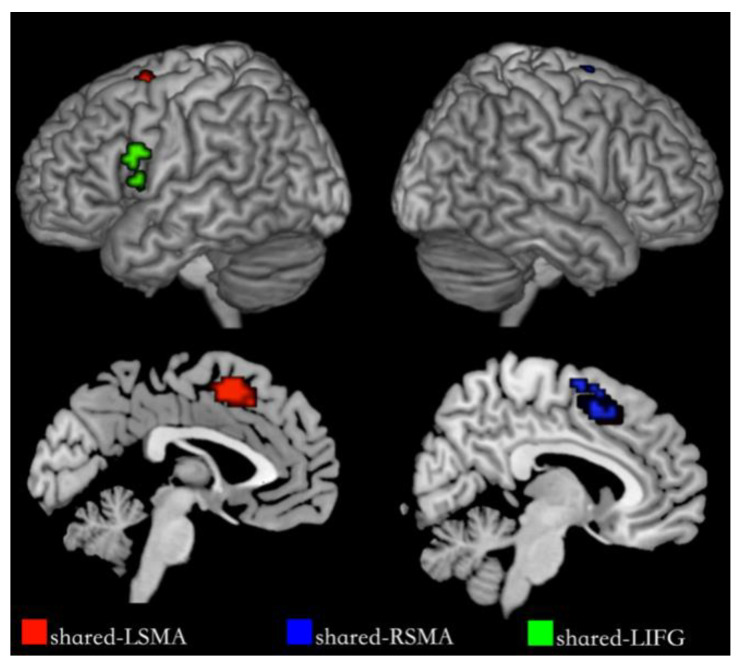
Shared regions for NA and NV. The red regions are shared-LSMA, the blue regions are shared-RSMA and the green regions are shared-LIFG.

**Figure 6 behavsci-12-00243-f006:**
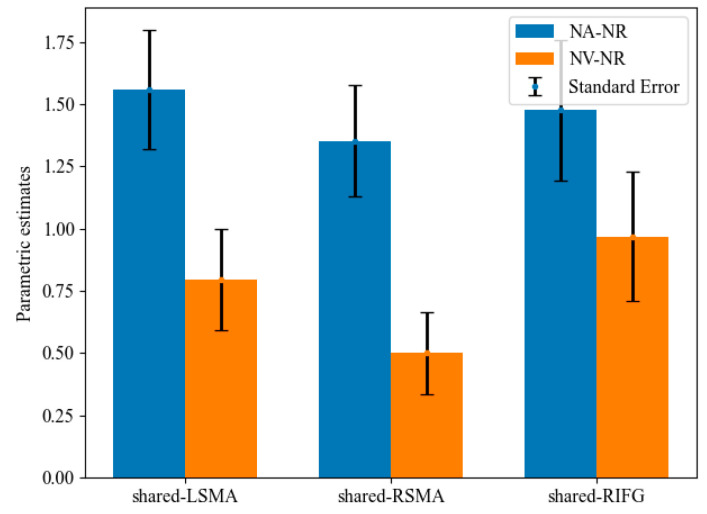
Parametric estimates for NA-NR and NV-NR in shared regions. The mean parameters of NA-NR were significantly higher than those of NV-NR in the shared-LSMA and shared-RSMA.

**Figure 7 behavsci-12-00243-f007:**
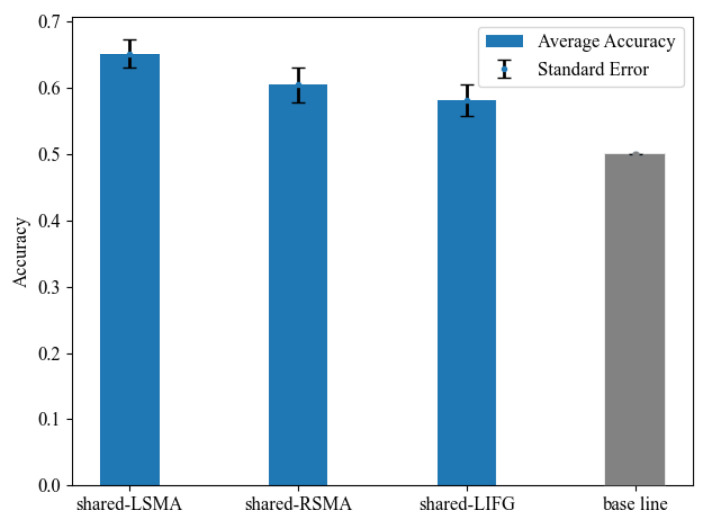
The classification accuracy and standard error in each shared region and the chance level of 50%. The accuracy in all ROIs was significantly above the chance level.

**Table 1 behavsci-12-00243-t001:** Activation peaks and extents for NA > NR.

NV > NR				MNI Coordinates		
Region Name	Brodmann Area	Voxel Count	*t*-Value	x	y	z
**bilateral Supplementary Motor Area,** **left Superior Frontal Gyrus**	6/32/4/40	793	7.25 5.81	−4 −18	8 1	51 57
**left Inferior Frontal Gyrus**	9/44	393	6.98	−45	5	9

**Table 2 behavsci-12-00243-t002:** Activation peaks and extents for NV > NR.

NV > NR				MNI Coordinates		
Region Name	Brodmann Area	Voxel Count	*t*-Value	x	y	z
**bilateral Supplementary Motor Area**	6/32	249	6.56	−14	1	66
**left Inferior Frontal Gyrus**	9/44/45	367	6.10	−38	22	9

**Table 3 behavsci-12-00243-t003:** Activation peaks and extents for NA > NV.

NA > NV				MNI Coordinates		
Region Name	Brodmann Area	Voxel Count	*t*-Value	x	y	z
**left Precentral Gyrus,** **bilateral Supplementary Motor Area,** **left Supramarginal Gyrus**	4/6/24/3	365	5.93 5.84 5.93	−35 −4 −52	−19 5 −23	54 48 54

**Table 4 behavsci-12-00243-t004:** Voxel clusters for the conjunction analysis of NA > NR and NV > NR.

		MNI Coordinates		
Region Name	Voxel Count	x	y	z
**left Inferior Frontal Gyrus**	29	−45	8	6
**bilateral Supplementary Motor Area**	116	−4	12	57

**Table 5 behavsci-12-00243-t005:** Results for MVPA between NA>NR and NV>NR.

ROI	Average Accuracy (%)	Accuracy SE	*t*(20)	*p* Value
**shared-LSMA**	65.21	0.021	7.35	<0.001
**shared-RSMA**	60.52	0.026	4.11	<0.001
**shared-LIFG**	58.18	0.024	3.25	0.002

## Data Availability

The data presented in this study are available on request from the corresponding author. The data are not publicly available due to privacy.

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
