# Peer review of "Different Neural Activities for Actions and Language within the Shared Brain Regions: Evidence from Action and Verb Generation"

_behavsci, 2022, doi:10.3390/bs12070243_

Round 1
Reviewer 1 Report
This fMRI study aims to assess the neural activity associations between action and verb generation to object nouns, submitting that action and verb generations tasks’ activation levels and activity patterns would significantly differ in the shared regions. In general, the topic and results of the study are of potential interest. However, this work presents a number of methodological concerns and various inaccuracies, which should be adequately addressed:
- The Authors have enrolled a small number of only 21 subjects. Therefore, the Authors should clarify how they have determined this number of subjects and whether it is optimal for the specific purpose of this study;
- Previous studies have emphasized the importance of characterizing the performance of scanners for fMRI (Friedman and Glover, J Magn Reson Imaging 2006, 23: 827-839), given the potential impact on results of clinical studies (Friedman and Glover, Neuroimage 2006, 33: 471-481). However, the Authors have mentioned no quality control for guaranteeing the correct functioning of their scanner. The Authors should hence adequately discuss this main limitation of the study;
- The Authors have not indicated the BW and echo spacing values used for fMRI acquisitions. In this regard, previous studies have proven that BW and especially echo spacing can substantially affect image quality (Giannelli et al, Med Phys 2010, 37: 303-310; Giannelli et al, J Appl Clin Med Phys 2010, 11: 3237). Therefore, the Authors should adequately discuss this issue. Moreover, they should explain how BW and echo spacing values were chosen and whether their impact on image quality was preliminary assessed;
- The Authors should clarify why they have used a suboptimal anisotropic acquisition voxel for fMRI acquisitions;
- The Authors should clarify whether they have used a gradient echo or spin echo EPI sequence. Moreover, they should clarify why no parallel imaging technique was employed to reduce EPI off-resonance artifacts;
- The Authors have not reported at all the acquisition parameters of T1-weighted acquisition sequence;
- The description of how structural MRI images were coregistered to BOLD images and segmented to grey matter, white matter and CSF is insufficient;
- Functional images were smoothed using a kernel of 8 mm FWHM. The Authors should discuss whether this kernel is optimal and how it can bias their results;
- The slow signal drift was eliminated by using a 1/128 Hz cutoff high-pass filter. The Authors should clarify how they chose this cutoff value and whether it is optimal;
- The threshold value of activation maps was p < 0.001 at the voxel level. I suppose that the Authors have applied this p value without any correction for multiple comparisons. The Authors should discuss whether and how this choice can bias the results.
Reviewer 2 Report
The manuscript entitled “Different neural activities for action and language within the shared brain regions: evidence from action and verb generation” presents a very interesting study which, with the use of an fMRI experiment, explores the neural activity associations between actual actions and verb generation. Although I enjoyed reading this manuscript, and I definitely identify its contribution to knowledge, since I believe that it has something new to offer to scientists and health professionals, there are several issues which could be addressed in a revised version of the paper. I detail these in the comments below:
1. Introduction
· The introduction is well-written and contextualized with respect to previous empirical research on the topic. The last sentences of the first paragraph, though, refer to a conclusion based on previous studies which is not explained thoroughly, and it is not closely related to the present study (“In addition to the object nouns, motion evoked potential studies also found that the perception of abstract words could activate hand-related regions [7]. These results have shown that not only object nouns or related verbs, but also other types of language can affect the motor system”). Therefore, if the authors decide to keep these sentences, they should explain why the perception of abstract words activates hand-related regions.
· Dorsal premotor cortex results to dPMC and not to PMCd
· It is not clear what the abbreviation “GLM” stands for. [… in the shared region were similar in general linear model (GLM)].
2. Materials and methods
· It would be interesting if the authors could provide the readers with some information regarding participants’ years of education.
· The term “noun images” gives the impression that the authors used pictures, something which is not true. Therefore, they should consider the use of “written words of the nouns”.
· Why there were only 11 participants who took part in the rating study? Did they have similar characteristics to the ones of the participants of the main experiment?
· The sentence “The action consistency scores were the highest number of feedback of an object divided by the second-highest feedback” is not clear. More elaboration is needed.
· Figure 1: again the use of the term “noun pictures” is misleading since they are not actual pictures but written words. It is not clear to me why the word “envelope” is related to “pinching” and not, for example, the word “needle”, In general, the categorization of the nouns is not clear. Was there any difference regarding participants’ performance in the different groups of nouns?
· One would expect the noun reading task to be the first participants would have to perform, so as to be a real control task without any priming effect. Could the authors explain when participants were asked to perform this task and why?
· It is not clear why “After the MRI scanning, participants were asked to recall the actions and verbs they had made to the nouns”. Authors should explain the purpose of this specific task.
· Similarly, the authors should explain the 2.4. Behavioral Analysis.
3. Results
The presentation of the results is clear and informative
4. Discussion
The conclusions of the study should be clearer, as well as the connection between the results of the present study and the medical conditions of Parkinson’s disease and apraxia.
Lastly, the constraints of the study and the suggestions for future research are missing.
Reviewer 3 Report
The authors conducted a careful fMRI study comparing common object (noun) and action (verb) naming. They found greater IPL, PcG, SMA and PMC activity during action naming than verb naming. The dissociations they found are partially concordant with prior literature, but also open up new directions for research (mostly through their noun reading analysis, in contrast to previous work eliciting noun productions).
First, I think the authors should connect their results to psycholinguistic models of action and object processing. There is a slew of research that is relevant here, and I think the author’s results could contribute to some models of semantics in the brain.
Many researchers have claimed that the neural basis of complex event semantics (actions, verbs) is widely distributed, as the authors seem to suggest (Murphy 2020, Ohlerth et al. 2021).
References:
Murphy, E. (2020). The Oscillatory Nature of Language. Cambridge: Cambridge University Press. https://doi.org/10.1017/9781108864466.
Ohlerth A-K, Bastiaanse R, Negwer C, Sollmann N, Schramm S, Schröder A and Krieg SM (2021) Benefit of Action Naming Over Object Naming for Visualization of Subcortical Language Pathways in Navigated Transcranial Magnetic Stimulation-Based Diffusion Tensor Imaging-Fiber Tracking. Front. Hum. Neurosci. 15:748274. doi: 10.3389/fnhum.2021.748274
Reviewer 4 Report
This study compared between action and verb generation with affordance-driven object nouns. This is done because similar activation level have been reported in the literature using GLM, and thus the authors aimed to use MVPA to test whether the GLM findings would truly hold at the pattern level.
The authors used three conditions (NV (verb generation), NA (action), and NR (reading control)) for comparisons in this study, which looks good to me. But the literature review could have been better as much of the neurophysiological work is not covered yet.
Given that the authors have done 3 types of action movements, I was wondering if the authors have also tried to analyze possible overlap/difference between the 3 movements?
Also, a block design was used, is this typical for MVPA studies? Or are there other reasons to why this was done? The possible confounding effects of participants’ “task set” is usually more prominent in block designs since they can adopt a strategy or state of mind and just keep going.
Minor point, there were several instances where the authors mentioned using “pictures” as stimuli, but I think all stimuli were words, correct?
Reviewer 5 Report
Authors compared the differences and similarities of brain regions activated by action and verb generation using task-based fMRI. They found that the action generation task activated more in the dorsal PMC and the left IPL than the verbal generation task. They also found that bilateral SMA and
left IFG were shared by action and verb generation.
Method is clearly written and can be replicated by other researchers. The results are clearly presented so as discussion.
Only minor concern is that they employed 11 participants to decide which nouns to be used. It might cause some bias, so this can be limitation for the study. Including this, I would like the authors to mention the limitation of the study.
Round 2
Reviewer 1 Report
I am sorry to say that the Authors have not adequately addressed the main critical comments raised in my previous review. The Authors have not discussed and recognized the methodological limitations that I have indicated. In particular, they have uncritically and simplistically cited other studies which presents the same limitations. In order to improve the manuscript, I hope the Authors are able and willing to reconsider my constructive comments.
Reviewer 4 Report
I don't quite understand why an analysis between the action types can't be done (or attempted)... this could be done by aggregating the trials, so the block design should not be a problem.
Round 3
Reviewer 1 Report
I thank the Authors for adequately addressing my previous critical comments.
Author Response
Thank you again for your criticism and suggestions. We will pay attention to these crucial limitations in future studies.
Reviewer 4 Report
I appreciate the authors' response, though even a block design can support a simple contrast between the conditions? This would map the relevant regions for grasping, pressing, and punching. Thanks.
Author Response
We are very sorry, maybe our last explanation was not clear enough. The key reason for our difficulty in analyzing the three movements is that we did not record the occurrence time of the corresponding noun trial for each movement in the experiment. In our experiment, a block might have random nouns for different types of movements with random orders. For example, in an action generation block, after the order of shown nouns could be:
Handbag, Hammer, Clip, Stapler, Button, Clamp
The first and second nouns are associated with grasping. The third and sixth nouns are associated with pressing. The fourth and fifth nouns are associated with pinching. However, in the experiment, we have not recorded the orders of nouns in each block. We just record the orders of blocks in the experiment. For example:
NA block, NR block, NV block, NR block.....
The orders of blocks were also randomized. The reason we did this is that in the design of the experiment, we only considered the difference between comparing action and verb generation, and we didn't consider the difference between comparing the three types of movements. Therefore, now we only know the types of blocks corresponding to fMRI timeseries. However, we don’t know the nouns for movement types corresponding to fMRI timeseries.
The conditions we recorded were just NA, NV and NR blocks, but not the specific nouns in each block.